# Safety Management and Wellbeing during COVID-19: A Pilot Study in the Manufactory Sector

**DOI:** 10.3390/ijerph19073981

**Published:** 2022-03-27

**Authors:** Gloria Guidetti, Michela Cortini, Stefania Fantinelli, Teresa Di Fiore, Teresa Galanti

**Affiliations:** 1Department of Psychological, Health and Territorial Sciences, University G. d’Annunzio of Chieti-Pescara, 66100 Chieti, Italy; gloria.guidetti@unich.it (G.G.); michela.cortini@unich.it (M.C.); teresa.difiore@unich.it (T.D.F.); 2Department of Humanities, Literature, Cultural Heritage, Education Sciences, University of Foggia, 71100 Foggia, Italy; stefania.fantinelli@unifg.it

**Keywords:** COVID-19 pandemic, wellbeing, safety management, manufacturing sector

## Abstract

Background: The rapid spread of COVID-19 has generated anxiety and concerns among the whole population, by also affecting people’s working life quality. Although several studies underlined the impact of the COVID-19 pandemic in the healthcare sector, very few studies investigated the consequences in the occupational sectors with low risk of contagion. Method: 220 full-time in-presence workers of the manufacturing sector agreed to participate in a study of cross-sectional design during September and October 2020. Data were collected by means of a self-reported questionnaire conceived to investigate the constructs of the COVID-19 concerns, both the personal contribution and the supervisor support to workplace safety, the organizational commitment to safety, and finally, the level of workers’ exhaustion. Results: This study highlights that COVID-19 concerns represent a significant source of stress since it is significantly associated to higher levels of exhaustion among workers. Furthermore, the findings show the relevance of resources related to employee’s personal contribution to safety management as well as the role of climate variables. Conclusions: These results promote knowledge on the role of COVID-19 concerns in affecting psychological wellbeing at work, as well as the impact of both individual and job-related resources that may prevent exhaustion at work. Finally, the present findings also have implications for organizations and the maintenance of their commitment to safety.

## 1. Introduction

The COVID-19 outbreak has radically impacted on work processes and organizations, by putting health and safety management at the fore. Organizations typically implement an occupational health and safety management system to control risks and ensure a safe working environment and optimal health for their workers. During the COVID-19 pandemic, however, employees are facing new psychosocial risks in the workplace and every organization has had to place more emphasis on workplace safety management practices to mitigate risks and safeguard health and wellbeing [1,2]. In the field of work and organizational psychology, several studies have been conducted to understand the pandemic’s impact on workers’ wellbeing and mental health and to underline the transformations of the organizational world [3,4]. Some of these studies tried to underline the consequences of the new ways of working adopted by the organizations to counter the spread of contagion, such tele-working and smart working [5,6,7]. In particular, there are studies that investigated how job activities and familiar context can be successfully matched or not [8]. Another research line has referred to the healthcare sector, more exposed at risk of contagion and experiencing high levels of stress and psychological disease [9,10,11]. On the contrary, very little has been said about those occupational sectors, such as manufacturing businesses and industries, that have returned to work in presence, with no other choice. Indeed, based on level of exposure, proximity, and aggregation, manufacturing industries are considered at low occupational risk of infection by SARS-CoV-2 [12]. Despite this, it is equally important to investigate health and safety issues among those sectors that nevertheless had to face a significant reorganization of work processes to ensure a safe return to work after the interruption of working activities during the first emergency phase [13]. Moreover, as Sinclair [14] and Weissberg [15] argued, workplace can be a source of transmission and many jobs can be potentially hazardous by fostering employees fears and concerns related to COVID-19 exposure. If, on the one hand, COVID-19 concerns have a relevant role in affecting stress and psychological discomfort [16,17], resources that may help to alleviate the negative effect of stress situations by promoting organizational factors of safety climate and by involving workers in safety solutions should be addressed [14,18,19]. In relation to this, in addition to engineering, administrative, and PPE measures that are mainly used to counteract the COVID-19 contagion [20], it is paramount to understand the role of psychosocial factors and of those “non-technical skills” that can affect both the application of technical measures and the perceived psychological safety and wellbeing of workers [21,22,23].

According to this, the aim of this study was to respond to the need to extend research on COVID-19 safety issues and wellbeing among a working population in the manufacturing sector that compared to other working categories has received scant attention so far. To this end, the results will promote knowledge on the role of COVID-19 concerns in affecting psychological wellbeing even among “low risk contagion” occupational sectors, as well as the impact of both individual and job-related resources that, by supporting workers’ safety, may prevent exhaustion when facing pandemic stress in the workplace. Furthermore, as the total exit from the pandemic situation is a rather long journey, in respect of which behavior and safety measures cannot be dismissed, these findings may be informative for those figures directly involved in occupational health and safety management during the transition to the so-called “new normal”.

### Literature Review and Hypothesis Development

The rapid spread of COVID-19 worldwide has generated anxiety and concern among the entire population [16,24] and also affected people’s quality of working life [25,26]. Especially among essential employees in social and health services, it has been highlighted that the feeling of fear of being infected and worries related to pandemic consequences play a significant role in influencing mental health and negative emotional contagion [27,28,29], increasing symptoms of anxiety, depression, stress, and burnout [30,31]. It has also been shown that concerns about COVID-19 are a source of stress and discomfort even for those who have undertaken smart working, fostering a feeling of social isolation and decreasing productivity and job satisfaction [6]. Therefore, even among employees in those jobs considered essential especially for the economic recovery, which, unlike tele-workers, have returned to work while increasing their exposure to infection risk [22,32], COVID-19-related concerns may represent a source of exhaustion and reduce mental health. Since negative emotions may decrease safety behaviors and encourage counter-productive ones [33], it is relevant to understand how this source of stress may also negatively affect the psychological wellbeing of manufacturing workers who returned to work during the pandemic crisis, to prevent future negative outcomes and promote a safer work environment.

According to these assumptions, we state the following hypothesis:

**Hypothesis** **1** **(H1).**
*COVID-19 concerns are positively associated with exhaustion.*


Along with the role played by COVID-19 concerns as a relevant stressor, the way by which organizations respond to the emergency can have significant effect on health, safety, and employee wellbeing. Along with all the other solutions adopted to combat COVID-19, such as the provision of sanitizers, the adoption of social distance, the creation of new work shifts, and the disinfection of workplaces, that resulted in improvements in safety and health performance [32], proper risk management and communication can create a sense of awareness and appropriateness of the measures. According to this, scholars have emphasized the application, already contained among the main directives issued for the management of health and safety in the workplace (e.g., EU OSH framework Directive 89/391/EEC; Italian Legislative Decree 81/08), of the integrated and participatory approach in risk assessment and management during the pandemic emergency [12,13]. Through participatory actions, risk management must therefore involve, in addition to the figures responsible for safety and prevention in the workplace, the active participation of employees who are responsible agents for managing their own and others’ health. Specifically, it has recently been recognized that a “total worker health” approach to occupational safety should be applied [13]. These authors emphasized that the key elements to protect safety, health, and wellbeing during COVID-19 pandemic relate to leadership commitment, supportive working conditions, active employee participation, and comprehensive and collaborative strategies.

Within the pre-pandemic literature on workplace safety, the role of aspect characterizing the safety climate, which indicates the specific system of perceptions related to organizational safety arrangements, has been extensively outlined [34]. In this regard, the safety climate can be characterized by multiple sources, such as perception of safety training, informal processes at the group level by means of social support, leadership best practices, and organizational safety management [35,36]. Past research has highlighted how safety and health climates are linked to a range of factors, including safety practices, motivation, knowledge, and accidents, as well as physical health and wellbeing [37,38]. In addition, a growing stream of research focuses on the role of safety climate in maintaining positive health and wellbeing [39] as, according to Clarke [38], a negative perception of the safety climate is linked to stress and can reduce psychological wellbeing.

For the purposes of the present study, two main elements that characterize the safety climate will be considered: supervisor support and organizational commitment to safety. Related to the challenge of COVID-19, Sinclair and colleagues [14] argued that “the role of supervisors and managers in understanding and responding to occupational health, safety, and wellbeing concerns has never been so important” (p. 11). Indeed, since supervisor support has been identified as one of the most powerful resources to prevent burnout [40] and even to promote safety at work [41], its role should be further elevated during periods of high-stress conditions constituted by the emergency period. In this sense, as evidenced among university staff, supervisors represent a source of support to prevent exhaustion and uncertainty due to the COVID-19 crisis [23,42]. Therefore, this role should also be explored among workers who returned to work after the lockdown for whom the supervisor may be a relevant source of information regarding safety and stress management programs, policies, and resources. 

Above and beyond the support experienced in the relationship with a direct supervisor, more distal correlates of the safety climate should be considered. Brown and colleagues [43], applying a “total worker health approach” [13], evidenced in this regard that, among SME workers, perceptions of organizational commitment to safety and health climate were the strongest predictors of self-reported wellbeing during the first wave of COVID-19. Furthermore, an interesting finding evidenced from this study is that these aspects proved to be more predictive compared to perceptions of organizational response to COVID-19 and work and life changes during the pandemic. This means that a strong safety culture constitutes a key element in the management of emergency and crisis situations.

Finally, pre-pandemic literature has also emphasized the role of individual factors that are inherent to behavior, motivation, and commitment toward safety [44,45]. Instead, studies suggest that safety at work should be understood as a process in which all components interact with each other and contribute to generate a healthy and safe workplace [46,47]. Workers’ behaviors aimed at helping co-workers, making suggestions for change, and promoting safety programs constitute key aspects of safety participation in the workplace [48]. Within this line of research, recently, it has been proposed a tool aimed at measuring the perception of safety at work during the pandemic, by integrating the measurement of the employee’s personal contribution to healthy and safe work practices in relation to COVID-19 [21]. According to the integrated and participatory approach that has been emphasized to protect safety, health, and wellbeing during COVID-19 pandemic [12,13], individual contributions to safety can therefore be evaluated in relation to wellbeing, alongside elements of the safety climate. All these aspects may therefore be conceptualized as personal and job-related resources that, according to Rudolph and colleagues [18], should be integrated within the research on employee health and safety during the pandemic by contributing to alleviate the high-stress situation induced by the pandemic.

According to these assumptions, we state the following hypotheses:

**Hypothesis** **2** **(H2).**
*Personal contribution to workplace safety in relation to COVID-19 is negatively associated to exhaustion.*


**Hypothesis** **3** **(H3).**
*Among safety climate perceptions:*

*supervisor support to workplace safety in relation to COVID-19 (H3a) is negatively associated to exhaustion*

*organizational commitment to safety (H3b) is negatively associated to exhaustion.*


## 2. Materials and Methods

### 2.1. Participants and Procedure

The present pilot study on the quality of working life and safety perception during the COVID-19 pandemic was performed in a multinational factory. A cross-sectional design was used to collect data by means of a self-reported questionnaire using Qualtrics platform during September and October 2020. At the time of data collection, all the participants were full-time, in-presence workers, namely, no work-from-home workers were involved in the survey. The research conforms to the Declaration of Helsinki of 1995 (and following revisions), and all ethical guidelines were followed as required for conducting human research, including adherence to legal requirements of the studied country. According to the country legislation, this study did not contemplate an ethics approval, as there were no special procedures or treatments that could be source of stress for participants.

Moreover, participants volunteered for the research without receiving any reward, signed the informed consent, and agreed to anonymously complete the questionnaire.

Out of 550 employees invited to participate in the research, 220 agreed to respond to the survey, and 152 questionnaires were correctly filled out and used for the present study. Of those, a total of 29 employees were females (19.1%) and 123 were males (80.9%), with a mean age of 44.4 (SD = 8.3, min = 24, max = 61) and a mean length of service of 14.28 years (SD = 9.3). Regarding job role, 94 (61.8%) were blue-collar workers, whereas 58 (38.2%) were administrative employees.

### 2.2. Measures

COVID-19 concerns were assessed with 4 items developed by Toscano and Zappalà [6] (e.g., Coronavirus worries me). The scale ranged from 1 (totally disagree) to 5 (totally agree). 

Personal contribution to workplace safety in relation to COVID-19 was assessed with 4 items developed by Converso and colleagues [21] (e.g., Think about your feelings regarding your job in this phase. Do You feel able to provide information to other employees regarding how to tackle contagion risks by COVID-19). The scale ranged from 1 (not at all) to 5 (completely).

Supervisor support to workplace safety in relation to COVID-19 was assessed with 2 items specifically developed for this study and aimed at evaluating the perceived support experienced in the relationship with direct supervisors (e.g., My supervisor has been supportive in this phase of COVID-19 emergency). The scale ranged from 1 (not at all) to 10 (completely).

Organizational commitment to safety was assessed with 3 items adapted from Taylor and Snyder [48] (e.g., The organization considers worker safety a very important thing). The scale ranged from 1 (not at all) to 5 (completely).

*Exhaustion* was assessed with the 8-item scale of the Oldenburg Burnout Inventory [49] (e.g., After my work, I regularly feel worn out and weary). The scale ranged from 1 (strongly disagree) to 5 (strongly agree).

### 2.3. Data Analysis

Data analyses were performed using IBM SPSS, Version 26 (SPSS Statistics for Windows, Version 27.0., IBM Corp, Armonk, NY, USA), Version 27), and Mplus 8 (Computer Software, Los Angeles, CA, USA). Confirmatory factor analysis (CFA) was used to assess the dimensionality of the scales, and internal consistency was assessed by the Cronbach’s alpha coefficient for each synthetic index. Multiple regression models were performed to evaluate the association between COVID-19 concerns and job resources of safety measures in relation to emotional exhaustion. A hierarchical regression process (enter method) was run to evaluate the marginal utility of variables entered at each step. Control variables, namely, gender, age and role, were entered in the final step.

## 3. Results

### 3.1. Preliminary and Descriptive Statistics

First, two CFAs were conducted to compare a five-factor model, which considers all the study variables as distinct, with a model in which all the items were grouped into a single dimension using an MLR estimator. The five-factor model showed a greater fit to data (χ^2^ = 282.47; df = 179; χ^2^/df = 1.57; comparative fit index (CFI) = 0.92; Tucker–Lewis index (TLI) = 0.90; root mean square error of approximation (RMSEA) = 0.06, 95% CI = 0.05–0.07; standardized root mean square residual (SRMR) = 0.07) compared to the model with a single factor grouping all the items (χ^2^ = 888.20; df = 189; χ^2^/df = 4.70; CFI = 0.40; TLI = 0.34; RMSEA = 0.15, 95% CI = 0.14–0.16; SRMR = 0.16). The fit values of the five-factor model were good, and each item loaded into its factor with saturation values greater than 0.40. 

Pearson’s correlations between the study variables’ means, standard deviations, and Cronbach’s alphas are presented in Table 1.

The scales and subscales had adequate internal consistency, and all the variables correlated in the expected direction with emotional exhaustion; however, COVID-19 concerns did not significantly correlate with any of the other variables except for age, evidencing a positive correlation.

### 3.2. Regression Analysis

The results of the multiple regression analysis are shown in Table 2.

In the first step, COVID-19 concerns were entered, evidencing a positive association with exhaustion (β = 0.26; *p* < 0.001). In the second step, personal contribution showed a significant association with exhaustion (β = −0.23; *p* < 0.001). In the third and fourth steps, supervisor support (β = −0.22; *p* < 0.01) and organizational commitment to safety (β = −0.21; *p* < 0.05) were inserted and evidenced, respectively, a significant and negative association with exhaustion. In the last step, control variables were added, and specifically age, gender, and role; however, no significant relationships were evidenced regarding any of the control variables.

A significant increase in R^2^ was observed after adding the step two variables, and when adding the variables in the firth and fourth steps. No significant increasing change in R^2^ was observed after inserting control variables.

## 4. Discussion

The present study aimed to shed a light on safety and wellbeing during the COVID-19 pandemic emergency among workers of the manufacturing industry. Despite this sector being among those that suddenly returned to work after the closures imposed by the first lockdown, it has received scarce attention. To achieve that goal, it was analyzed how psychological wellbeing at work can be influenced, on the one hand, by negative emotions related to pandemic emergency, i.e., COVID-19 concerns. On the other hand, it was analyzed how resources related to the safety climate in the workplace and safety participation can positively affect psychological wellbeing by reducing levels of exhaustion.

The results of the analyses showed that COVID-19 concerns represent a significant source of stress and psychological discomfort since they are significantly associated to higher levels of exhaustion among workers. These results are in line with past evidence that has demonstrated the association between COVID-19 concerns and negative wellbeing and organizational outcomes [6,16,23,27] among other occupations The present findings, however, allow to highlight that this source of stress and discomfort due to pandemic emergency can have a significant impact on wellbeing levels even among those workers employed in occupational sectors defined as “low risk of contagion” who, at the same time, had to face the return to work in the middle of the pandemic without having specific knowledge on the management of biological hazards such as COVID-19. In this vein, organization can implement targeted interventions by recognizing the role exerted from the psychological dimension of risk, in addition to objective analyses of risk exposure.

Furthermore, the role of resources related to safety climate and safety participation has emerged, also evidencing a significant increase in the variance explained in each subsequent step of the regression analysis. Specifically, personal contribution to workplace safety in relation to COVID-19 was significantly and negatively associated to exhaustion. Similarly, both supervisor support in relation to COVID-19 and organizational commitment to safety were significantly and negatively associated to exhaustion. Therefore, the hypotheses (H2, H3a, and H3b) were all supported, and it is interestingly to note that after including safety climate variables, i.e., supervisor support in relation to COVID-19 and organizational commitment to safety, the relationship between personal contribution and exhaustion decreased. This could mean that, in addition to the role of personal contribution, the role played by safety resources in the workplace proved to be further significant and relevant in preventing psychological exhaustion during the stressful phase of returning to work.

On the one hand, concerning the role of each employee’s personal contribution to safety management, the present findings are consistent with the pre-pandemic literature that has paid increasing attention to the role of proactivity and safety behavior in promoting and maintaining an adequate safety culture and wellbeing in organizations [45,46]. Moreover, these results enhance more recent evidence on the role that employees can proactively play in promoting a safer work environment in times of COVID-19 [21,23] by also suggesting that organizations should sustain or promote such positive behaviors.

In relation to the above statement, by focusing on aspects relating to the safety climate variables, the present study contributed to knowledge of how organizations can deal with the return-to-work phase by ensuring safety and productivity. Specifically, with regard to the role of supervisor’s support in relation to COVID-19 safety management, the present findings are in line with recent research that has emphasized the role of this resource in other occupational context [23] (or among those that, differently from the manufacturing sector, have continued working from home [42]). Finally, these results are in line with the work of Brown and colleagues [43], who, among SME enterprises, highlighted the fundamental role of organizational commitment to safety in promoting psychological wellbeing in times of pandemic crisis. We can therefore argue that the results of the present study further integrate the recent evidence, highlighting that both the presence of an organizational commitment anchored to a solid culture of safety and of resources specifically activated during the COVID-19 pandemic and based on supportive relationships with direct superiors are necessary when employees face atypical stress due to pandemic emergencies. According to the EU strategic framework on health and safety at work (2021–2027), the COVID-19 pandemic has shown the strategic role played by OSH in the protection of workers’ health and, at the same time, the continuity of economic and social activities [50]. In this sense, to reactivate the productivity, it seems crucial that organizations invest in the promotion of a risk-assessment process and in the diffusion of a safety culture where all employees play their part to ensure occupational health and safety [51].

While these results may contribute to the development of research by also having practical implications, certain limitations should also be considered. First, the cross-sectional, nonrandomized nature of the study design and the data collection among a single multinational only limit the generalizability of the results. In this vein, future studies should enlarge this research scope to other manufacturing realities to test the hypothesis on a larger sample size. Furthermore, longitudinal studies should be applied in order to monitor how organizations adapt to the changes and different phases of the pandemic and to consider any reciprocal relationships between variables using cross-lagged analyses.

## 5. Conclusions

Since COVID-19 has accelerated and amplified changes in the nature and management of work, creating unexpected challenges for safety, health, and wellbeing, this study contributes to bridging a knowledge gap currently affecting occupational sectors less exposed to contagion but equally involved in safety issues. Therefore, practical implications can be derived with regard to preventive measures and health promotion practices. On the one hand, in order to counteract stress and psychological discomfort during actual pandemic periods, organizations should monitor, in addition to the physical risk, the psychosocial correlates of concerns related to COVID-19 exposure. In this vein, organizations may implement training interventions aimed at educating employees on COVID-19 psycho-physical correlates along with stress-management interventions that can enhance personal resources in facing COVID-19 concerns and fears of contagion.

Furthermore, organizations could intervene by promoting and safeguarding the participatory actions in risk management, fostering, through adequate interventions such as safety and health training, each individual employee’s personal contribution to COVID-19 safety, for example, through interventions that can sustain employee’s motivation and proactivity behavior to cope with COVID-19 at the workplace. This intention is in line with a proactive attitude in the health and safety management [47,48] and with the philosophy of continuous improvement, as professed by the Japanese *Kaizen* in the manufacturing sector [52]. Kaizen is meant as a set of strategies aimed at nurturing sharing and a climate of positive synergies, requiring effort and participation from all individuals and levels in organizations [52].

Given the significant role exerted from supervisor support in managing COVID-19 safety at the workplace, organizations should also provide specific training to supervisors or leaders on how to support workers in times of crisis, fostering their critical role in safety management and supporting employees when faced with concerns and threat [53]. In this vein, specific training should be provided aimed at consolidating and keeping up-to-date knowledge about COVID-19 and its implications on psycho-physical health, as well as stress management practices that on the other side can help support the supervisor to navigate these difficult tasks. Moreover, managers should promote employees’ participation in safety by providing them with opportunities to discuss with each other and consulting with them, to ensure employees’ cooperation in COVID-19 diffusion [2]. Finally, the present findings also have implications for organizations and the maintenance of their commitment to safety. Indeed, as highlighted by Brown and colleagues [43], companies that are committed to employee safety are also likely to have strong leadership support and use of safety and health procedures to protect employees from risks. Therefore, this suggests that organizations that nurture their commitment to safety may be better prepared to maintain employee wellbeing during unexpected times of crisis and emergencies.

The results of the present study can be regarded in the light of the EU’s 2021–2027 Occupational Health and Safety (OSH) framework [50], which focuses on a risk-prevention culture and lays down employers’ obligations on (i) risk assessments, (ii) preventive measures, (iii) giving OSH information to workers, (iv) training, (v) consultation, and (vi) balanced participation. However, beyond a risk prevention focus, it could be highlighting a health promotion focus that, by leveraging employees and organizational resources, can sustain healthy and resilient organization processes [54] that could be of further value in coping with the challenges of the COVID-19 age.

## Figures and Tables

**Table 1 ijerph-19-03981-t001:** Correlations among the study variables, descriptive and reliability statistics.

	1	2	3	4	5	6
1. Exhaustion	1	0.28 **	−0.29 **	−0.33 **	−0.28 **	−0.02
2. COVID-19 concerns		1	−0.13	−0.12	−0.08	0.17 **
3. Personal contribution			1	0.43 **	0.11	0.72
4. Supervisor support to workplace safety				1	0.21 *	−0.02
5. Organizational commitment to safety					1	0.11
6. Age	−0.02	0.17 *	0.07	−0.02	0.11	1
Mean (SD)	17.4 (4.7)	12.2 (3.6)	15.3 (3.4)	15.07 (4.2)	13.4 (2.2)	44.9 (8.3)
Cronbach’s α	0.79	0.71	0.83	0.90	0.89	nn

* *p* < 0.05; ** *p* < 0.01.

**Table 2 ijerph-19-03981-t002:** Regression parameters: standardized coefficients and overall changes in R^2^ for exhaustion.

		Beta	Standard Errors
1			
	COVID-19 concerns	0.26 ***	0.10
	ADjR^2^ = 0.06		
	ΔR^2^ = 0.07		
	R^2^ = 0.07		
2			
	COVID-19 concerns	0.23 **	0.10
	Personal contribution	−0.26 ***	0.11
	ADjR^2^ = 0.13		
	ΔR^2^ = 0.07		
	R^2^ = 0.14		
3			
	COVID-19 concerns	0.22 **	0.09
	Personal contribution	−0.17 *	0.11
	Social support	−0.22 **	0.21
	ADjR^2^ = 0.16		
	ΔR^2^ = 0.04		
	R^2^ = 0.18		
4			
	COVID-19 concerns	0.21 **	0.09
	Personal contribution	−0.17 *	0.11
	Social support	−0.18 ***	0.22
	Organizational commitment to safety	−0.21 *	0.16
	ADjR^2^ = 0.21		
	ΔR^2^ = 0.05		
	R^2^ = 0.23		
	COVID-19 concerns	0.22 **	0.09
	Personal contribution	−16 *	0.11
	Social support	−0.18 *	0.09
	Organizational commitment to safety	−0.21 **	0.16
	Age	−0.04	0.04
	Gender (0 = male)	−0.06	0.9
	Role (0 = blue collar)	−0.07	0.7
	ADjR^2^ = 0.20		
	ΔR^2^ = 0.01		
	R^2^ = 0.23		

* *p* < 0.05; ** *p* < 0.01; *** *p* < 0.001.

## Data Availability

The data presented in this study are available on request from the corresponding author with the permission of the manufacturing company involved in the study. The data are not publicly available due to third party restrictions.

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
