# Peer review of "Safety Management and Wellbeing during COVID-19: A Pilot Study in the Manufactory Sector"

_ijerph, 2022, doi:10.3390/ijerph19073981_

Round 1
Reviewer 1 Report
The article concerns an interesting and at the same time important issue, which is the issue of COVID 19 in the workplace. As it is known from the transmission practice, this virus has a negative impact not only on the health of employees, but also its impact on the functioning of the enterprises.
My insights to be developed:
- The authors on lines 14, 42-43 indicate occupational sectors with a low risk of contagion. It would be worth indicating what these sectors are and what criteria have been adopted in order to classify a given sector as a sector with a low risk of contagion. It is also worth justifying why the manufacturing sector was included in this category;
- line 30 – the Authors should specify how the COVID-19 influenced the management of occupational health and safety;
- line 58 - why the Authors of the study indicate that the population working in the manufacturing sector is neglected? - To what extent is this group neglected? Why is this group neglected? Who has this population been neglected by?;
- descriptions in tables 1 and 2 – the Authors should enter table headings;
- line 254 - should be R2;
- the references should be revised according to the IJERPH guidelines.
I hope that my insights will be useful for the Authors of the study.
Reviewer 2 Report
- The authors need to explain why they have applied two scales (1-5 and 1-10) in the paper regarding scales used.
- Chronbach's alpha is missing for variable 6 in Table 1. Please, add it.
- I suggest the authors include a figure to show the relationship among H1, H2, and H3.
- I propose the authors enrich their comments in the Discussion section by including some data, especially in the second paragraph of this section.
Additional comments:
- What is the main question addressed by the research?
The main question addressed in this paper is: How has COVID-19 impacted the low-risk contagion occupational sectors, especially among the neglected working population in the manufacturing industry?
- Do you consider the topic original or relevant in the field, and if so, why?
I consider the topic relevant in the field, given the few studies dealing with this topic in the literature. They have mainly focused on the pandemic’s psychological, social, and economic consequences in the most exposed economic sectors and industries.
- What does it add to the subject area compared with other published material?
The literature published on this topic has dealt mainly with health-related, psychological, and economic issues, not with the quality of working life, workplace safety, and safety perception linked to COVID-19 in a multinational factory.
- What specific improvements could the authors consider regarding the methodology?
Data has been obtained in one multinational only, so data results and interpretation are limited to this case. I suggest the authors increase their scope of research for future studies.
Some additional comments on data results (CFI, TLI, RMSEA, and SRMR) obtained in the “3.1. Descriptive statistics and correlations” (page 5) should be welcomed. I recommend enlarging the Discussion section by commenting on results obtained with the specific data obtained. This section is relatively poor, and more practical information can be added from the results obtained.
- Are the conclusions consistent with the evidence and arguments presented, and do they address the main question posed?
Conclusions are slightly vague. When writing about practical implications in the firm, I suggest the authors be more specific and detailed. For example, from an economic perspective, I miss applying Kaizen-related strategies to create positive synergies in the firm or adopting business strategies linked to EFQM. Also, regarding workplace safety in firms, the EU’s occupational safety and health (OSH) legislation should be cited in the Discussion section and the conclusions. In this sense, I suggest the authors analyze the 2021-2027 EU strategic framework on health and safety at work report.
- Are the references appropriate?
References are appropriate, sufficient, and updated—no further comments on this issue.
- Please include any additional comments on the tables and figures.
The information regarding the betas (Table 2, page 6) is confusing. I suggest the authors explain carefully each statistical result obtained.
